# Inflammatory Immune Process and Depression-like Behavior in Hypothyroid Rats: A [^18^F] DPA-714 Micro Positron Emission Tomography Study

**DOI:** 10.3390/ph16020279

**Published:** 2023-02-13

**Authors:** Yizhen Wang, Aijuan Tian, Fang Zhang, Jing Yu, Jianer Ling

**Affiliations:** 1Second Affiliated Hospital, Dalian Medical University, Dalian 116027, China; 2Nuclear Medicine Division, Shengli Oilfield Central Hospital, Dongying 257000, China; 3Hwa Mei Hospital, University of Chinese Academy of Sciences, Ningbo 315010, China

**Keywords:** hypothyroidism, depressive behavior, forced swimming test, tail suspension test, inflammatory immune

## Abstract

Hypothyroidism is closely related to mental disorders, mainly depression, through an as-yet-unknown mechanism. The cerebral inflammatory immune process has been implied to play a pivotal role in the onset of affective symptoms in several conditions. In order to gain insight into the mechanism underlying the depressive behaviors in hypothyroid rats, brain microglial activation was evaluated using micro positron emission tomography imaging with a translocator protein (TSPO) radioligand. Hypothyroidism was induced in adult male Wistar rats by administration of 0.05% propylthiouracil in drinking water for five weeks. Open field, forced swimming and tail suspension tests were employed to evaluate the depressive behavior in hypothyroid rats, and the relationship between the behavioral changes and brain microglial activation was evaluated using [^18^F] DPA-714 micro positron emission tomography imaging. The open field test revealed significantly reduced first-minute activity and rearing behavior in the hypothyroid group, as well as significantly increased immobility in the forced swimming test and the tail suspension test. Hypothyroidism induced significantly increased microglial activation in the hippocampus. The radioligand uptake in the hippocampus correlated negatively with first-minute activity in the open field test (*p* < 0.05), and the radioligand uptake in the hippocampus correlated positively with changes in the immobility time in the forced swimming test and the tail suspension test (*p* < 0.05). Immunohistochemistry also confirmed the activation of microglia and inflammatory bodies in hypothyroid rats. The results indicate that hypothyroidism can induce depressive behavior in adult Wistar rats, microglial activation in the hippocampus plays an important role in the depressive behavior in hypothyroid rats and the inflammatory immune mechanism may underlie the behavioral abnormalities in thyroid dysfunction. Furthermore, the findings in the present study suggest there might be a common mechanism underlying depressive behavior in adult-onset hypothyroidism and depression.

## 1. Introduction

Thyroid hormones (THs) are pivotal for the natural development of the central nervous system and the maintenance of its structural and functional integrity [1]. A deficiency of thyroid hormones may lead to a series of somatic, neuropsychological, and psychiatric symptoms, including psychomotor slowing, fatigue, memory and learning impairment, inability to concentrate and emotional disorders. Among the presentation in hypothyroid patients, the most known is probably cognitive impairment and depressive symptoms [2], which is usually explained in the literature as being related to a global or regional reduction in cerebral blood flow, metabolism, or dysfunction of neurotransmitter and TH receptors [3,4]. The mechanisms underlying the affective disorders in thyroid dysfunction remain unclear. It has been suggested that depression and the depressive symptoms in hypothyroidism may share an underlying brain abnormality, or may be causally related.

There is evidence that neuroinflammation has a close relationship with mental disorders in several conditions including treated HIV-positive individuals [5], sports-related brain injuries [6] and T-lymphotropic virus type 1 (HTLV-1)-associated myelopathy [7], as well as thyroid dysfunction. Chronic activation of brain glial cells has therefore been implied to be a major contributing factor for the mental disorders accompanying thyroid dysfunctions. The 18-kDa translocator protein (TSPO), which is also known as the peripheral benzodiazepine receptor (PBR), is highly expressed in the mitochondria of microglia, astrocytes and macrophages in the human brain. Increased expression of TSPO in microglia has been observed during the inflammatory process in several conditions. Cerebral microglial activation can therefore be assessed in vivo with TSPO radioligand binding in PET imaging [8].

[^11^C] PK11195 is a first-generation TSPO radioligand used in PET studies with numerous limitations, including relatively low binding specificity and poor signal-to-noise ratio, which lead to discrepant results [9]. The second-generation TSPO radioligand has a higher affinity for TSPO and a higher displaceable binding fraction, which is determined by the rs6971 single nucleotide polymorphism (SNP) of the TSPO gene, and therefore enables an accurate quantitative analysis of TSPO PET data. In a recent [^11^C] PBR28 PET study in HIV-positive patients, TSPO binding in the hippocampus, amygdala and thalamus was shown to associate with defected cognitive performance in tasks evaluating word and visual memory [5]. In this study, a pyrazolopyrimidine-based TSPO imaging ligand, N,N-diethyl-2-(2-(4-(2-18F-fluoroethoxy) phenyl)-5,7-dimethylpyrazolo[1,5-a] pyrimidin-3-yl) acetamide ([^18^F] DPA-714), was employed as a new probe for the detection of TSPO level in hypothyroid rodents. Previous studies illustrated the feasibility of [^18^F] DPA-714 for the detection of cerebral TSPO expression, which was more suitable for quantitative assay of TSPO expression in vivo [10].

The aim of this study was to test for microglial activation in hypothyroid rats and the correlation between TSPO expression measured by [^18^F] DPA-714 and depressive behaviors.

## 2. Results

### 2.1. Body Weight

One-way repeated measures ANOVA yielded a main effect of time [F (5,90) = 1104.90, *p* < 0.001], a main effect of group [F (1,18) = 19.14, *p* < 0.001], and significant group by time interaction [*p* < 0.001]. The differences in body weights between the two groups were not significant at baseline or the first week during the experimental procedure (262.1 ± 10.4 g for CON vs. 264.2 ± 10.5 g for HYPO; 291.4 ± 10.7 g for CON vs. 279.5 ± 16.8 g for HYPO). The body weight of the HYPO group was significantly lower than the CON group from the second week until the end of the experimental procedure.

### 2.2. Serum Thyroid Hormone Levels

Total T_3_ and T_4_ levels were not significantly different between the CON and HYPO rats at baseline. TT_3_ level was 0.24 ± 0.02 nmol/L and TT_4_ level was undetectable in the HYPO rats after 5 weeks of PTU treatment, while TT_3_ level was 1.86 ± 0.14 nmol/L and TT_4_ level was 99.68 ± 7.41 nmol/L in the CON rats at that time. Compared with the CON rats, significant decreases in TT_3_ and TT_4_ were observed in the HYPO rats after 5 weeks of PTU treatment (t = 11.45, *p* < 0.001 and t = 13.45, *p* < 0.001, respectively).

### 2.3. Open Field Test

At baseline, no significant differences were found between the two groups in any index examined. After the 5 weeks of PTU administration, there was no difference between groups in either the total activity or defecation during the open-field test. Furthermore, control rats and hypothyroid rats did engage similarly in locomotor activity in the central squares and the peripheral squares of the open field arena. Compared to the control rats, the hypothyroid rats showed a significant decrease in the first-minute activity (t = 3.511, *p* < 0.01) and rearing (t = 2.632, *p* < 0.05) at week 5. In the HP group, there were significant differences in first-minute activity and rearing between baseline and week 5 (see Table 1).

### 2.4. Forced Swimming Test

In the FST, we measured the rats’ time spent immobile, which is considered to be depression-like behavior, as opposed to active swimming or other escape-related behavior, during the second 5 min real test session. At baseline, there were no significant differences between the two groups in both the floating time and the climbing time. After the 5 weeks of PTU treatment, the immobility time of the hypothyroid rats (163.4 ± 3.9 s) was significantly longer than that of the control rats (116.8 ± 4.8 s) (t = 7.524, *p* < 0.001) (Figure 1), and the climbing time of the hypothyroid rats (78.9 ± 5.3 s) was significantly shorter than that of the control rats (99.7 ± 5.3 s) (t = 3.329, *p* < 0.01). In the HYPO group, there were significant differences in immobility time and the climbing time between baseline and week 5.

### 2.5. Tail Suspension Test

Figure 2 shows the immobility time during the TST. At baseline, there were no significant differences between the two groups in the immobility time. After the 5 weeks of PTU treatment, the immobility time of the hypothyroid rats (140.0 ± 5.3 s) was significantly longer than that of the control rats (102.1 ± 4.7 s) (t = 5.353, *p* < 0.001). In the HP group, there were significant differences in immobility time between baseline and week 5.

### 2.6. Changes in Hippocampus DPA-714 Uptake Induced by Hypothyroidism

The relative increase in DPA-714 uptake in ID/CC was significant in the hippocampus (0.171 ± 0.029 for CON vs. 0.212 ± 0.035 for HYPO, *p* < 0.05). According to the Pearson’s correlation test results, the radioligand uptake in the hippocampus correlated negatively with first-minute activity in the open field test (*p* < 0.05), and the radioligand uptake in the hippocampus correlated positively with changes in the immobility time in the forced swimming test and the tail suspension test (*p* < 0.05). One-way repeated measure ANOVA for hippocampus regions yielded a main effect of time, a main effect of group and a significant group by time interaction (all *p* < 0.001). Figure 3 shows DPA-714 uptake in ID/CC in the hippocampus over the 30 min following injection. Figure 4 shows the MR template used in the normalization of the rat brain images and typical transversal [^18^F] DPA-714 images of a control and a hypothyroid rat.

### 2.7. Immunohistochemistry

Figure 5 and Figure 6 show that the levels of TSPO, Iba-1 and NLRP3 in the hippocampus of the HYPO group were increased significantly compared with that of the CON group by independent-samples *t*-test (t= 3.085, *p* < 0.05, t = 3.300, *p* < 0.05 and t = 2.812, *p* < 0.05, respectively).

## 3. Discussion

Thyroid hormones are essential for the development and function of the central nervous system, especially in the regulation of the development and differentiation of neurons and neuroglia [11]. THs deficiency is closely associated with both neurological and psychiatric manifestations, including mental retardation, ataxia, psychosis, depression, mania, apathy and cognitive impairment. Overt hypothyroidism has long been reported to induce depression-like behavior in patients as well as rodents, and thyroid hormones have been used as accelerators or augments of antidepressant drugs for several decades. THs are also important to the proper function of the immune system [12]. Diverse findings have been reported about the proinflammatory activities of monocytes and macrophages during hypothyroidism. De Vito et al. reported increased levels of proinflammatory molecules such as macrophage inflammatory protein-1α and interleukin-1β as a result of the enhancement of phagocytosis and increased levels of reactive oxygen species (ROS) [13]. Conversely, hypothyroidism was shown to produce opposite effects on the immune system, leading to decreased immune response, antibody production, cell migration and antioxidant enzyme [14]. Microglia, as well as other glial cells in the CNS, play an important role in the relationship between thyroid dysfunction and neuropsychological disorders. Microglia are generally considered to be the primary immune cells of the CNS. Various pathological processes will activate microglia, trigger them to migrate toward the lesion site and begin phagocytosis of the damaged cells, and levels of TRα1 and TRβ1 have been identified in primary cultured rat microglia [15]. Microglial activation has been considered to contribute to various pathologies, including schizophrenia, depression and Alzheimer’s disease, as proved by recent positron emission tomography and postmortem studies.

Our study showed that hypothyroidism led to increased depression-like behaviors in adult male Wistar rats, as well as increased microglia and TSPO expression in the hippocampus, Moreover, the expression of TSPO in the hippocampus is related to the depressive behavior of hypothyroid rats. At the same time, we also detected an increase in inflammatory bodies in the hippocampus after hypothyroidism. This is similar to the results of several studies on the neuroinflammatory theory of depression. The study of Yalin Wang et al. showed that after 12 weeks of receiving chronic mill stress (CMS) in adult male SD rats, [^18^F] DPA-714 micro PET showed a significant increase in the uptake of radioactivity in the hippocampus, at the same time they also detected an increase in NLRP3 inflammasome and some inflammatory mediators in the hippocampus after chronic stress and found that minocycline treatment can alleviate the depressive- and anxiety-like behavior and neuro-inflammation induced by chronic stress [16]. The data from the present study added new proofs that oxidative stress induced by immunoinflammation played an important role in depressive behaviors in hypothyroid rats. This is not a unique instance—in another study, DPA-714 uptake increased in 11 regions of interest in the brain of depressed rats [17]. This suggests that depression-like behaviors associated with hypothyroidism in rats may have a common mechanism with depression.

Microglial activation has been considered to be related to neuropsychological abnormalities in several conditions. In a recent TSPO PET study, neuro-psychiatric and cognitive impairments in treated HIV-positive patients were shown to be correlated closely to cerebral inflammation. Microglial activation, as measured by TSPO uptake in PET imaging, was greatest in the subcortical grey matter, especially the basal ganglia [5]. Biomarkers of microbial translocation correlated with increased TSPO binding, leading to the postulation that microbial translocation might be a causative factor of cerebral inflammation in treated HIV-positive patients. Another study among National Football League (NFL) players who experienced repeated sports-related concussive and sub-concussive brain injuries revealed prolonged immune activation following years of exercise and competition. [^11^C] DPA-713 TSPO imaging showed increased microglial activation in the NFL players in eight brain regions including the supramarginal gyri, mesial temporal lobes and left temporal pole. Although none of the players had developed neuropsychological abnormalities at that time, there is evidence that NFL players are more likely to develop cognitive, affective and behavioral abnormalities later in life, which might be the result of microglial activation [6]. Further [^11^C] PBR28 studies demonstrated a close correlation between the severity of cognitive defect and cerebral inflammation in patients with human T-lymphotropic virus type 1 (HTLV-1)-associated myelopathy, in which the pathogenesis was thought to be a bystander effect of the cellular immune response to HTLV-1 infected lymphocytes that entered the central nervous system [7]. In such a context, microglial activation might be a promising therapeutic target for preventative interventions before any clinical manifestations. This hypothesis was supported by a ^11^C-(R)-PK11195 PET study in multiple sclerosis (MS) [18,19,20]. The TSPO radioligand uptake was increased in regions that corresponded to T2 lesions on MRI, which could be partially reversed by six months of fingolimod treatment. The results suggested the potential of anti-inflammatory drugs in the prevention of neuropsychological and cognitive impairment and in monitoring response to treatment in MS.

The changes in the ambulation mode of animals in the open field test (including the total activity, the central activity, the peripheral activity and the first-minute activity) are often used to reflect the emotional state in novel surroundings. Although the total lines crossed in the open field arena can indicate locomotion activity, a decrease in ambulation over time is also commonly interpreted as a measure of habituation.

The effects of hypothyroidism on locomotion are related to the rats’ age when hypothyroidism is induced. Hyperactivity has been consistently reported when the induction of hypothyroidism happened during the developmental period of the central nervous system (including prenatal, perinatal and neonatal hypothyroidism) [18,19], while reduced internal and external open-field activity was usually observed in adult-onset hypothyroidism [21]. The results in the present study were consistent with the previously reported studies—both central and peripheral activity decreased compared to the control group, although not significantly, which might be due to the differences in the experimental methodology.

In our study, the first-minute activity in the open field test was found to correlate negatively with the deactivation of brain activity in the hippocampus. The first-minute activity was mainly used to reflect emotion and motivation, reduced first-minute activity found in the present study revealed hesitation and decreased motivation to explore of the animals in novel surroundings. Our study also showed decreased rearing in adult-onset hypothyroidism, which was consistent with our previous studies [22]. The number of lines crossed and the frequency of rearing are usually considered indexes of locomotor activity, while they are also indexes of exploration and anxiety. The low frequency of such behaviors indicates hypoactivity or reduced exploration, as well as a higher level of anxiety. It is generally believed that the pattern of locomotion is less vulnerable in the adult period, and the decrease in locomotion observed in adult-onset hypothyroidism may be the result of reduced exploration and increased anxiety [21]. Control rats also showed a reduction in locomotion 5 weeks later, which could be explained as a sign of habituation to novel surroundings, although the difference was not significant compared to the baseline.

In our study, there was a significant increase in immobility time during both the forced swimming test and the tail suspension test in hypothyroid rats compared to control rats. The forced swimming test and the tail suspension test were developed as animal models of depression and could be used to evaluate the effectiveness of antidepressants [23]. The immobility time in the forced swimming test is considered a measure of behavioral response to an inescapable stress. Changes in immobility time in the forced swimming test are related to the severity of hypothyroidism, the methodology used to induce hypothyroidism and the species of animals used. In most studies, immobility time was reported to increase significantly in hypothyroid rodents with great sensitivity, even being 90% higher in thyroidectomized Wistar rats with severe hypothyroidism than in control rats [24]. Mild hypothyroidism could be induced in Wistar rats after 14 days’ iodine-free diet, and the immobility time was observed to increase by 60%. Clinical or subclinical hypothyroidism induced by total or hemi-thyroid electrocauterization in Wistar rats was reported to be associated with increased immobility in both FST and TST [25], which was consistent with the behavioral results of the present study. Our findings contrast with those of Jefferys and Funder who observed that chronic PTU administration for 21 days reduced immobile time significantly [6] and increased cortical T3 levels 9.5-fold in Sprague–Dawley rats [26], which reflects interstrain differences between these rat strains. In agreement with findings that severe affective and behavioral dysfunction were associated with hippocampal dysfunction, the results of the Pearson correlation test in this study showed that the depression-like behavior parameters, including the first-minute activity in OFT and the immobility time in FST and TST, were all closely related to changes in hippocampal TSPO expression in the micro-PET imaging, implying the importance of immuno-inflammatory mechanism in the induction of depression-like behavior in hypothyroid rodents.

The normal function of the hippocampus relies on thyroid hormone during development as well as in adulthood [27]. Adult-onset hypothyroidism could impair long-term memory, and the morphological and functional defects of the hippocampus were significantly related to depressive-like behavior, which could be reversed by thyroxin replacement therapy [28]. Even mild hippocampal impairment in SCH patients was reported to cause depression and memory deficits [29]. In our study, Pearson’s correlation test revealed a significant correlation between the first-minute activity in the OFT and the immobility time in the FST and TST and the changes in TSPO expression in the micro-PET imaging, indicating the important role of cerebral inflammation in the depressive behavior in hypothyroid rodents. This was consistent with results from mice models with cerebral astrocyte-specific Dio2 inactivation and reduced T3 level, which exhibited anxiety-depression-like behavior in the OFT, FST and TST [30]. Microarray gene expression profiling of the Astro-D2KO mice also revealed the enrichment of three gene sets related to inflammation and immune response and the impoverishment of three gene sets related to mitochondrial function and response to oxidative stress. The same study also revealed a reduction in mRNA levels of genes known to be affected in classical animal models of depression, which together with the findings in the present study that microglial cells were highly activated in the brains of hypothyroid rats and its relationship with the behavioral changes, implies the role of inflammation in the mechanism of depressive-like behaviors in hypothyroid animals.

In humans, the quantitative interpretation of the TSPO PET imaging results is confused by significant interindividual uncertainty in the binding affinity of these radioligands [31]. The binding class of these radioligands, therefore, needs to be ascertained before any quantitative comparisons of TSPO expression. It has been reported that the binding affinity of DPA-714, as well as other second generation TSPO radioligands, relates closely to the genotype of TSPO rs6971 polymorphism [32]. The binding class of these radioligands in the brain can thus be predicted simply by genotyping the TSPO. Caution should therefore be used to use high affinity rs6971 polymorphism in future tests in the demonstration of microglial activation in hypothyroid humans.

Our study is the first to apply TSPO PET to the exploration of the mechanism of hypothyroidism leading to depressive behavior (as far as we know). This may provide a new target for the treatment of depression-like behavior related to hypothyroidism in the future. But there are still some areas for improvement in our research. Firstly, we have not verified whether drug treatment (such as L-T4) can reverse the performance of rats in behavioral experiments. Secondly, although the role of OFT, TST and FST in exploring the mechanism of depressive behavior has been proved by numerous studies, we can also add motor analysis tests or sucrose preference experiments in future studies to distinguish the depressive behavior and dyskinesia associated with hypothyroidism.

## 4. Materials and Methods

### 4.1. Animals

Twenty healthy adult male Wistar rats (about 8 weeks of age) were used in this study. All rats were randomly divided into two groups and housed two per cage. Animals were raised at a room temperature of 22 ± 1 °C and a humidity of 50–60% with regular light–dark cycles (lights on at 8 AM and off at 8 PM).

During the experimental period, rats in the hypothyroid group (HYPO group) were supplied on ad libitum access to 0.05% propylthiouracil (PTU) dissolved in water for 5 weeks, and the same amount of water was provided to the control rats (CON group). PTU is an antithyroid drug that inhibits T_4_ deiodination in peripheral tissues. The dose of PTU was chosen according to several literature data which showed a significant reduction in serum T_3_ level and lowering of cellular thyroid hormone activity in both cerebral and peripheral tissues after 5 weeks of 0.05% PTU treatment [27]. All rats received the same amount of laboratory food. All animals were acclimatized to the experimental environment for more than one week prior to the test.

The behavioral experiment was conducted in the following order for 4 consecutive days: OFT(1 day)-FST(2 days)-TST(1 day), This arrangement follows previous research. Such an arrangement can minimize the impact on the results caused by sequence problems. After PET scanning of all rats, euthanasia will be carried out.

### 4.2. Verification of Hypothyroidism

All rats from both the CON group and the HYPO group were euthanized at the end of the study. Blood samples were collected by cardiac puncture and immediately centrifuged at 4000 rpm for 7 min. The plasma was separated and stored at −20 °C for further use. The total thyroxine (TT_4_) and total 3,5,3′–triiodothyronine (TT_3_) levels were analyzed via a radioimmunoassay procedure. All radioimmunoassay kits were purchased from 3V Bioengineering Group Limited, China.

### 4.3. Open-Field Test

The open field test is conducted by imitating the method of C.M. Poveda, et al. [33]. The open field apparatus was a 100 × 100 × 40 cm^3^ wooden box with a black square base and four black walls. The base of the box was divided into twenty-five squares by white lines, including nine central squares and sixteen peripheral squares. The animals were not disturbed for more than 12 h before the test, and the procedure was performed under dim light (about 80 lux) in a quiet room beginning from 10 a.m. Each rat starts from the outer square of the field, and a video camera was mounted 1 m above the box. The rats’ behavior was recorded by the camera for further analysis, which was performed separately by two experienced technicians who were blind to the rats’ groups. The rats were allowed to move freely for 5 min in the box, and the following behaviors were recognized and analyzed: the number of squares crossed in the peripheral section, the number of squares crossed in the central section, the total activity during the 5 min test (including the central and peripheral squares crossed in the box), the first min activity, times of defecation, the grooming behaviors (combing, licking, washing, etc.), and the rearing behavior (animal standing vertically on its hind legs). The arena was completely cleaned with a 75% ethanol solution before each test to erase the rat’s smell.

### 4.4. Forced Swimming Test

The FST was conducted according to the protocol of R. Yankelevitch-Yahav, et al. [34]. Rats were subjected to the forced swimming test at the end of drug administration. A glass cylinder (20 cm in diameter) was used and filled up with 30 cm deep water so that the hind paws of the rat could not reach the bottom part of the cylinder. A pretest of 15 min was arranged before the real test. After the pretest, the rats were removed from the cylinder, dried and placed into their home cage. A real test was performed for 5 min 24 h later in the same cylinder. The experiment was recorded by video for further analysis of their behaviors. The time that the rat remained immobile was quantified during a test period of 5 min. Immobility was considered to be depression-like behavior. The total immobility time was defined as the whole amount of time when the rat remained immobile or made only small limb movements necessary to keep its head above the water surface.

### 4.5. Tail Suspension Test

The TST was performed according to the method of S.Q. Dong, et al. [35]. Each rat was suspended for six minutes by the tail using adhesive tape fixed approximately 1 cm from the end of the tail. After the first 2 min of the test, the total time of immobility (in seconds) was recorded. An animal was thought to be immobile only when it hung down passively and was completely motionless except for those movements required for respiration.

### 4.6. [^18^F] DPA-714 Micro PET Scan and Data Analysis

The micro PET/CT (PINGSENG Healthcare) was used for the brain imaging study. To monitor the immuno-inflammatory process in the rat brain by PET in vivo, [^18^F] DPA-714 dynamic PET scans were acquired 5 weeks after PTU administration. The injected dose was 1.96 ± 0.68 mCi (mean ± SD). Each rat was put on a heating pad at 30 °C for more than 30 min before the imaging process and until the end of the acquisition. Image acquisition was performed under isoflurane anesthesia (5% induction, 2–2.5% maintenance), and the head was fixed with tape to prevent accidental movements during the acquisition process. The image data were analyzed with Pmod 4.304, and the hippocampus was selected as the region of interest (ROI). Time activity curves (TACs) were generated based on the percentage injected dose per cubic centimeter (%ID/cc).

The dynamic PET acquisition was divided into one 5 s frame, five 10 s frames, one 35 s frame, three 60 s frames, one 180 s frame, and four 300 s frames for the total duration of the scan. The original data within each frame was binned into 3-dimensional sinograms, with a ring difference of 47 and a span of 3. Attenuation correction was accomplished with an attenuation map from the CT image. Scatter and attenuation corrections were performed, using a 2-dimensional ordered-subsets expectation-maximization algorithm with 16 subsets and 4 iterations. The sinograms were then reconstructed into tomographic images (128 × 128 × 95) with voxel sizes of 0.095 × 0.095 × 0.08 cm^3^. The CT image was first registered with the small-animal PET image, segmented, and then projected into sinogram space with a span of 47 and a ring difference of 23.

### 4.7. Immunohistochemistry

The rats were anesthetized with chloral hydrate (400 mg/kg) and then perfused transcranial with saline followed by 4% paraformaldehyde (PFA) (pH 7.4). For immunohistochemical labeling, rat brain tissues were fixed in 4% PFA and embedded in paraffin. Brain sections (5 μm thick) were prepared, deparaffinized and rehydrated in gradient ethanol. The slides were blocked with 10% goat serum in Phosphate Buffered Saline (PBS) (Abcam, Cambridge), and then incubated with the following primary antibody mixtures: anti-TSPO (1:200, affinity), anti-Iba-1 (1:200, Abcam), anti-NLRP3 (1:200, affinity) at 4 °C overnight. After being washed three times with PBS, the slices were incubated with goat anti-rabbit secondary antibody, washed again with PBS three times, stained with DAB (3,3′-diaminoben-zidine tetrahydrochloride) (Abcam, Cambridge) and counterstained with hematoxylin, and finally mounted, dehydrated, and covered with picks. Finally, the number of positive cells in three visual fields randomly selected by two people under a 40x light microscope was counted, and the average value was calculated for comparison between groups.

### 4.8. Statistical Analyses of Behavioral Tests

The data were analyzed using SPSS 15.0. Descriptive data were expressed as the mean (s) ± standard error of the mean (S.E.M.). Body weight and DPA-714 uptake were analyzed using a repeated measurement ANOVA with group as the between-subject factor and time as the within-subject factor. Trend analyses of the changes in %ID/cc of DPA-714 uptake and body weight in the hypothyroid group and the control group were performed with one-way repeated measure ANOVA. The least-square deviation (LSD) post hoc *t*-test was used for multiple comparisons between different time points. Student’s *t*-test was used for the comparison of means between the two groups. The relationship between the behavioral test and brain DPA-714 uptake was determined by Pearson correlation analysis. All tests were two-tailed. The significant level was set at *p* < 0.05.

## 5. Conclusions

This study showed that adult-onset hypothyroidism results in depressive behaviors in male Wistar rats, as well as microglial activation in the hippocampus. Behavioral changes correlated closely to TSPO radioligand uptake in the hippocampus. The results of the present study suggest that inflammatory immune processes underlie mental disorders in hypothyroid rats. There might be a common mechanism underlying depressive behaviors in hypothyroidism and depression.

## Figures and Tables

**Figure 1 pharmaceuticals-16-00279-f001:**
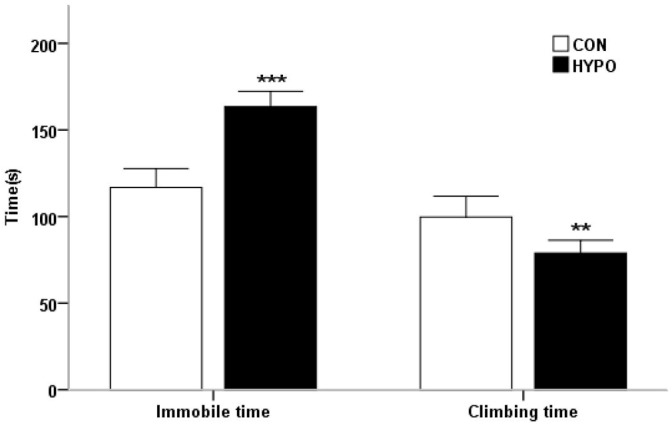
Behavior of the control group (n = 10) and hypothyroid group (n = 10) in the forced swimming test (FST) 5 weeks after PTU administration. Comparison of time spent climbing and time spent immobile. (** *p* < 0.01, *** *p* < 0.001).

**Figure 2 pharmaceuticals-16-00279-f002:**
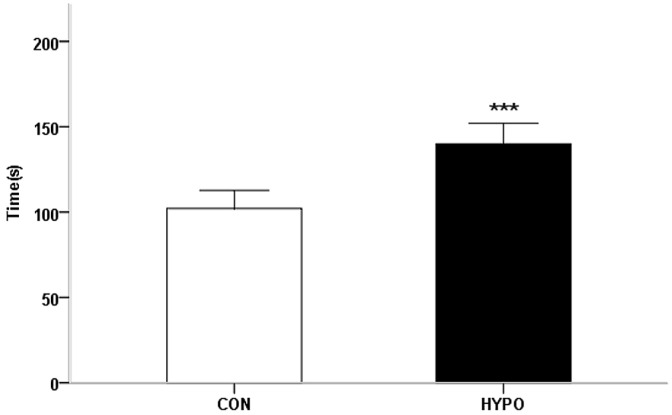
Behavior of the control group (n = 10) and hypothyroid group (n = 10) in the tail suspension test (TST) 5 weeks after PTU administration. Comparison of time spent immobile. (*** *p* < 0.001).

**Figure 3 pharmaceuticals-16-00279-f003:**
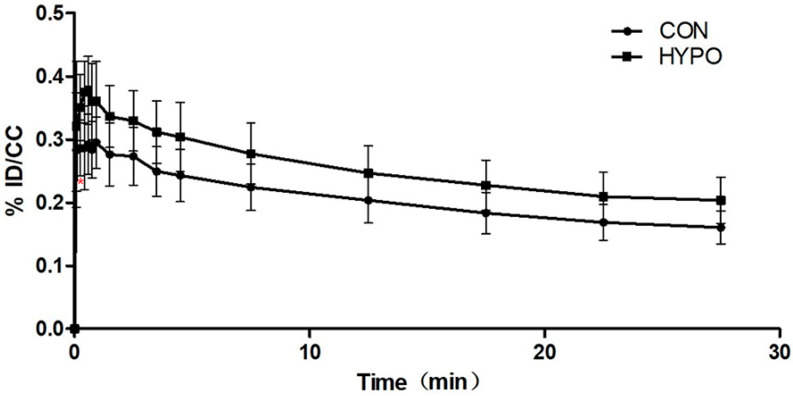
[^18^F] DPA-714 uptake over the 30 min following injection in the hippocampus. Repeated measurement ANOVA showed a significant difference between the two groups. Red * indicates a significant difference in DPA-714 uptake between the two groups after this time.

**Figure 4 pharmaceuticals-16-00279-f004:**
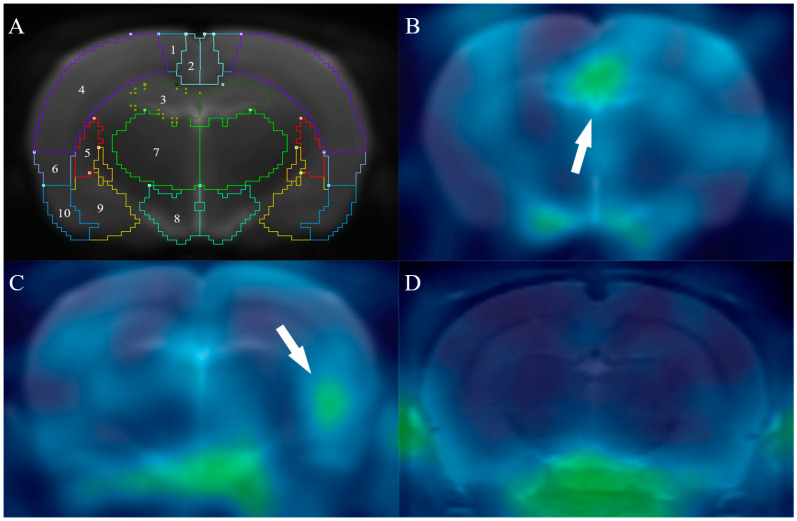
Typical cross-sectional rat brain images. The area indicated by the arrow is DPA-714 concentration area. (**A**) MR template for PET image standardization of rat brain (hippocampus section). 1. motor cortex, 2. retrosplenial cortex, 3. anterior hippocampus, 4. somatosensory cortex, 5. caudate putamen, 6. insular cortex, 7. thalamus, 8. hypothalamus, 9. amygdala, 10. entorhinal cortex. (**B**) Hippocampal uptake in a hypothyroid rat. (**C**) Uptake of the hippocampus at another section in a hypothyroid rat. (**D**) [^18^F] DPA-714 images of a control rat.

**Figure 5 pharmaceuticals-16-00279-f005:**
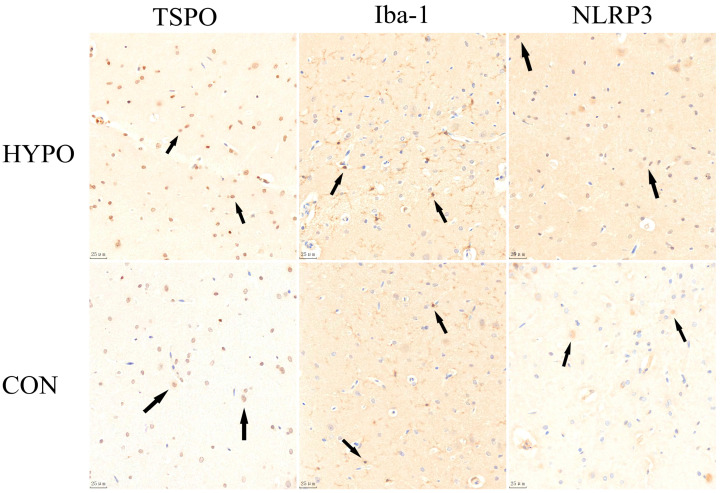
Immunohistochemical results of TSPO, Iba-1, NLRP3 expression (Under 40× optical microscope). Brown cells are positive (as shown by the black arrow) and blue cells are negative.

**Figure 6 pharmaceuticals-16-00279-f006:**
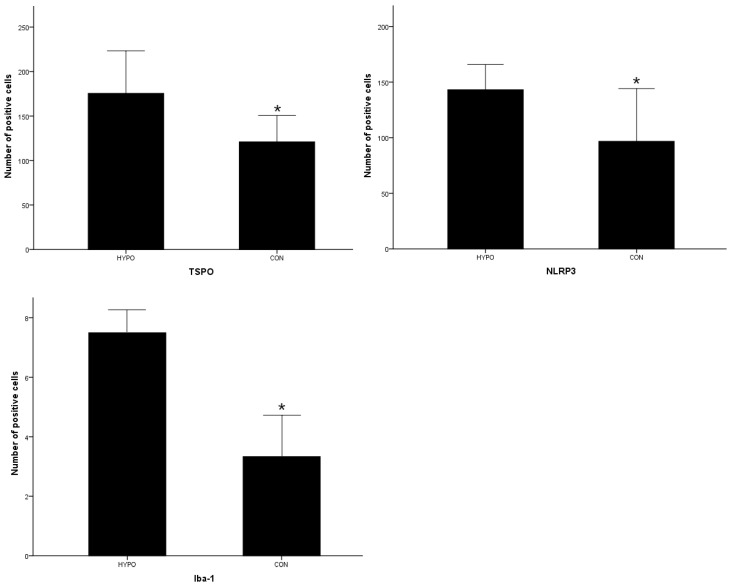
Quantitative analysis of immunohistochemistry positive cells (* *p* < 0.05).

**Table 1 pharmaceuticals-16-00279-t001:** Open field test results in CON rats and HYPO rats at baseline and week 5.

	CON		HYPO	
	Baseline	Week 5	Baseline	Week 5
Number of squares crossed				
Total	66.70 ± 2.96	60.10 ± 2.38	60.50 ± 2.76	62.00 ± 2.58
Central	12.90 ± 1.15	11.50 ± 0.78	11.30 ± 0.76	10.80 ± 0.66
Peripheral	53.80 ± 2.32	48.60 ± 2.10	49.20 ± 2.67	51.20 ± 2.84
M1	24.60 ± 1.08	23.40 ± 1.24	22.50 ± 0.99	17.40 ± 1.18 ^##ΔΔ^
Number of times				
Rearing	10.30 ± 0.56	9.50 ± 0.54	9.90 ± 0.64	7.60 ± 0.48 ^#Δ^
Grooming	3.20 ± 0.36	3.60 ± 0.22	3.80 ± 0.42	2.90 ± 0.38
Number of feces counted				
Defecation	7.90 ± 0.46	8.40 ± 0.43	8.90 ± 0.38	8.00 ± 0.39

Open field test results shown by CON rats (n = 10) and HYPO rats (n = 10) at baseline and week 5. Values are expressed as mean ± SD.^#^
*p* < 0.05, ^##^
*p* < 0.01 compared with baseline; ^Δ^
*p* < 0.05, ^ΔΔ^
*p* < 0.01 compared with CON group at week 5. M1: first-minute activity.

## Data Availability

Data is contained within the article.

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
