# Peer review of "Inflammatory Immune Process and Depression-like Behavior in Hypothyroid Rats: A [18F] DPA-714 Micro Positron Emission Tomography Study"

_pharmaceuticals, 2023, doi:10.3390/ph16020279_

Round 1

Reviewer 1 Report

Although the behavioral results demonstrated in the present study are known, as several models of hypothyroidism (induced by methimazole, propylthiouracil or thyroidectomy) have investigated locomotion, exploration, cognition, anxiety and depression-like behavior, there is an important mechanism exploration with the use of the [18F] DPA-714 Micro Positron Emission Tomography. In order to make the manuscript even more suitable I would like to suggest some questions:

1) In general text:

- Use italics for in vivo (lines 74 and 375) and ad libitum (line 322).

- p-value must be italicized and lower case whenever used.

- The commas in lines 159-161 are in different formatting. The same happens with hippocampus in line 381. In line 161, the comma is missing before respectively.

- Line 200-201: "unique instance, in another".

- Line 335: missing data from the radioimmunoassay kit manufacturer.

- Line 400: correct ABACAM.

2) In the Abstract and Introduction:

- The mechanisms involved in mental disorders triggered by hypothyroidism (or hyperthyroidism) are not unknown. The literature contains several experimental studies that related changes in serum levels of thyroid hormones with changes in synaptic pathways, levels and receptors of different neurotransmitters, damage to brain structures, oxidative stress, etc. Therefore, the present study adds a mechanism that can be related to those already described and these relationships should be explored.

3) In the Results:

- In Figure 3: is the difference significant at all points analyzed? Please insert the information in the graph highlighting at which points there is a difference.

- In Figure 4: - Insert an image of capture in the hippocampus of a control animal and an animal induced to hypothyroidism (captured at the same time) for comparison purposes. Also, at what time was image B captured? And image C: what does this capture represent? Was it seen only in hypothyroid animals?

- In Figure 5: Improve resolution or marking indication.

4) In the Discussion:

- There is no justification for using male animals. Hypothyroidism is more common in women, so why only male rats were used in the experimental model? Ideally, they should test both sexes or start the analysis with female rats.

- It was not clear the order in which the behavior tests were carried out and whether there was a break between them or at the end before tomography and euthanasia. These details can have a major effect on the results, as tests that explore depressive behavior are very stressful. Furthermore, it is not usual to subject rats to the tail suspension test (normally, we only use mice that have lower weight due to the risk of stress and tail loss), so it will be important to clarify why you made this choice.

- Lines 231-235: Assessing motor and exploratory activity does not only reflect the emotional state. Motor analysis tests are necessary to demonstrate that the animals are able to perform the other tests, for example, forced swimming. If the animal showed motor impairment, there would be no way to prove that the lack of activity was depression-like behavior or just motor impairment.

- Lines 245-259: An adequate assessment of anxiety would only be possible with the performance of a specific test, such as the elevated plus maze. Since you performed the open field and analyzed the peripheral and central crossings separately, the analysis of central crossings best correlates with anxiety-related behavior. Defecation was also unchanged, so there is nothing to relate to possible changes in anxiety-like behavior. Altered rearing alone has little strength to be considered an anxious behavior, being just a drop in exploratory behavior.

- Better explore the results obtained that demonstrate the activation of microglia with those already demonstrated in the literature, for example, oxidative stress.

5) In the Materials and Methods:

- There was no description of data from the ethics committee authorization in the text.

Reviewer 2 Report

In this Manuscript the Authors have analyzed the effect of hypothyroidism on the depressive behavior of adult Wistar rats. They found that the [18F] DPA-714 uptake by positron emission tomography imaging and immunohistochemical markers of microglial activation were both increased in the hippocampus of hypothyroid rats. These alterations correlated with the behavioral abnormalities induced by hypothyroidism. 
The article is interesting and their findings show a possible common mechanism linking hypothyroidism and depression.

I have the following comments:

1) Hypothyroidism can have an impact on muscle functions and locomotor system, leading to fatigue. In addition, hypothyroidism reduces body temperature which is associated with a reduced locomotor activity in rodents. I suggest to better explain in the discussion why the differences observed between the two groups, are due to an altered behavior and not to an altered locomotor activity. 

2) Figure 5: the figure is not enough explicatory, since it is difficult to appreciate differences in the expression of TSPO, Iba-1 and NLRP3. I suggest adding a panel with the quantification of positive cells. In addition, did the Authors find any differences in glial density/glial activation of other areas of the brain associated with depression (e.g anterior cingulate cortex, prefrontal cortex, and amygdala)?

3) I suggest to add a paragraph, acknowledging the limits and strengths of their study. Among the limitations, they have not tested if these alterations were reversible with L-T4 treatment.

Reviewer 3 Report

This is an interesting manuscript where the authors have tried to establish a correlation between depressive behavior in patients of adult-onset hypothyroidism through preclinical research. The authors have been able to show through functional imaging, the microglial activation in  hypothyroidism-induced model of rats. The manuscript is fit for publication in its present form. 

Author Response

Thank you for your approval of this study.